# Application of the Van Cittert Algorithm for Deconvolving Loss Features in X-ray Photoelectron Spectroscopy Spectra

**DOI:** 10.3390/ma17030763

**Published:** 2024-02-05

**Authors:** Giorgio Speranza

**Affiliations:** 1Fondazione Bruno Kessler, v. Sommarive 18, 38123 Trento, Italy; speranza@fbk.eu; 2Department of Industrial Engineering, University of Trento, v. Sommarive 9, 38123 Trento, Italy; 3IFN-CNR, CSMFO Laboratory, v. alla Cascata 56/C, 38123 Trento, Italy

**Keywords:** deconvolution, spectral interference, noise interference, van Cittert, iterative algorithm

## Abstract

The convolution of two physical entities, denoted as *f* and *g*, delineates the manner in which one entity undergoes modification in response to the other. This transformative process is mathematically represented by the expression *f* ⨂ *g*, symbolizing the convolution of the two entities in a resultant function *h*. Frequently, it becomes imperative to comprehend the magnitude of the induced modifications. From the derived function *h*, a crucial step involves the separation of the two original signals, a process commonly referred to as deconvolution. Various techniques have been proposed to facilitate the calculation of the deconvolution, with one notable approach originating in 1931 by van Cittert. The algorithm, based on an iterative method, has been scrutinized over time, notably by Bracewell and, more recently, by Jansson. This work represents the current state-of-the-art, focusing specifically on the analysis of Auger spectra obtained through XPS. Emphasis is placed on delineating the procedural aspects of the analysis, and the algorithm utilized in the open-source software *RxpsG* is comprehensively described.

## 1. Introduction

Interference among data poses a common challenge across various spectroscopic techniques. An illustrative instance is the signal overlap encountered in astronomy, necessitating the application of deblurring techniques for the removal of atmospheric effects from observed signals through deconvolution methods [1]. Similarly, deconvolution proves important in mitigating turbulence effects within data acquired from diverse engineering and geophysical systems, spanning a broad spectrum of spatial and temporal scales [2]. Although resolving such issues through numerical simulations which can describe local motion features is conceivable, the associated computational effort becomes prohibitive. Consequently, deconvolution techniques are employed to alleviate computational load while upholding a comprehensive system description. Another noteworthy application of deconvolution is the analysis of strain maps derived from materials subjected to mechanical testing [3]. Deconvolution is explored in this context to mitigate noise and recover the actual displacement and strain fields from localized digital image correlation maps or localized spectrum analyses. This underscores the pivotal role played by deconvolution in enhancing the fidelity of data interpretation and analysis in diverse scientific domains.

In addition to addressing noise, within the context of the current study, another prevalent scenario where deconvolution finds widespread application is in mitigating interference stemming from the contribution of distinct spectral lines. The challenge lies in the unknown intensity and position of these interfering lines, thereby complicating the reconstruction of the resultant spectrum [4]. Broadly, the experimental spectrum is seen as a convolution in the analysis, with the objective of enhancing resolution through deconvolution. This interpretation involves considering the experimental spectrum *h* as the convolution of the original line shape *f* with a function *g*. This function *g* represents the contribution of additional spectral lines, the source of spectral Gaussian broadening, or systematic instrumental broadening [5,6,7]. By adopting this perspective, the deconvolution process becomes integral to unraveling the complexities of the spectral data and discerning the underlying components contributing to the observed interference.

In this manuscript, we will illustrate the application of deconvolution in analyzing Auger spectra obtained through X-ray Photoelectron Spectroscopy (XPS). In XPS, X-photons are utilized to excite electrons to the vacuum level. The kinetic energy of the detected photoelectrons can be correlated with the orbital of the emitting atom, enabling speciation and quantification of element abundances. During photoemission, a valence electron may relax into the core hole created by X-excitation. The energy released can excite a second valence electron, known as the Auger electron, into the vacuum level.

The Auger spectra can be conceptualized as the self-convolution of two valence states: the first corresponds to the electron relaxing into the hole, and the second involves emission into the vacuum after absorbing the released energy. This process mixes all valence states, rendering the Auger spectrum as the self-convolution of the density of states (DOS). Consequently, analyzing Auger spectra offers valuable insights into the electronic structure of valence electrons, including spectral changes induced by chemical bonds [8,9,10]. Regrettably, the photoemission process is intricate, and the resulting spectra are the overlap of various processes. Core lines, originating from the direct absorption of an X-photon and emission from an inner level, and Auger spectra typically appear at well-separated kinetic energies. However, photoelectrons from core levels may lose part of their energy in exciting valence electrons, generating a loss features such as plasmons, shake-up, and shake-off structures [8,9]. As these features impact the electronic structure of valence electrons, Auger spectra are perturbed and cannot be acquired separately from the contributions due to their loss features. Because they affect the electronic structure of the valence electrons, their effect is convolved with the pure Auger spectrum.

So far, to obtain a clear description of the valence structure, it is necessary to deconvolve the effects of relaxation processes from the Auger spectrum. The scope of this manuscript is to provide a comprehensive overview of potential solutions for signal deconvolution. Deconvolving the loss features from the Auger spectrum will be used as an example to showcase the deconvolution process and highlight the solutions implemented in the *RxpsG* software (https://github.com/GSperanza accessed on 24 January 2024) for signal processing [11].

## 2. Analysis of the Auger Spectrum and the Van-Cittert Algorithm

The core–valence–valence Auger transitions involve the relaxation of a valence electron into a core hole state and transfer of the emitted energy to a second valence electron that is emitted as an Auger electron. The kinetic energy E_kin_ of the Auger electron can be described by the following expression:E_kin_ = E_i_ − E_l_ − E_m_ − U_eff_(1)

Here, E_i_ − E_l_ represents the energy released in the relaxation process from the valence electron *l* in the core-level *i*. E_m_ is the is the binding energy of the second valence level *m* required to bring the Auger electron to the vacuum level. Finally, U_eff_ represents the core–valence hole–hole interaction.

The Auger spectrum, being the self-convolution of the density of states (DOS), is described by Equation (1) for all (*l*, *m*) combinations. In the past, numerous studies [12,13,14] have highlighted the Auger spectra’s sensitivity to valence electronic and local structures.

An illustrative case is the Auger spectrum of graphite, extensively studied [15,16,17] as a model for aromatic systems to examine initial state and core hole screening effects. Theoretical calculations emphasize that core hole screening significantly alters the shape and magnitude of the measured DOS in graphite [17,18]. In addition, it is recognized that misinterpretation of the Auger spectra may arise from neglecting single-particle and collective interactions, leading to ionization losses and plasmon losses [19]. This issue complicates the analysis of chemical bonding effects on Auger spectra. The effects of these interactions can be correctly described only through an accurate analysis of the Auger line shape. Smith and Levenson’s work [15] serves as an early example, where data were processed as the first derivative of the Auger spectrum, followed by background subtraction and deconvolution of loss features. A first attempt to deconvolve loss features from MNN Auger transitions of indium was made by Mulaire and Pereira [19]. They applied the inverse of the Fast Fourier Transform (iFFT) to calculate the deconvolved Auger spectrum. Similar methods were employed by other authors [15]. Later on, another try was carried out by other authors [20] who tried to solve the problem in a more systematic approach.

If *h* represents the convolution of two signals *f*(*t*) and *g*(*t*), expressed as *f* ⨂ *g*,
(2)$$h\left(t\right)=f\;⨂\;g\;\int_{-\infty }^{+\infty }f\left(\tau \right)g\left(t-\tau \right)\;d\tau$$
all the authors of the cited works exploited the property that
FFT[*h*(*t*)] = FFT[*f* ⨂ *g*] = FFT[*f*(*t*)] · FFT[*g*(*t*)] (3)

It follows that the deconvolution of the function *g* from *h* is simply computed as
FFT[*f*(*t*)] = FFT[*h*(*t*)]/FFT[*g*(*t*)] (4)

The deconvolved spectrum *f* is easily obtained by taking the inverse of the FFT:*f*(*t*) = iFFT{ FFT[*h*(*t*)]/FFT[*g*(*t*)]} (5)

By applying Equation (5) to deconvolve the loss features from the Auger spectrum we obtain
C KVV_D_ = iFFT {FFT(C KVV_M_)/FFT(C 1s)} (6)
where C KVV_D_ represents the deconvolved Auger spectrum, C KVV_M_ corresponds to the measured Auger spectrum, and C 1s is the carbon 1s core line.

However, this approach does not clearly elucidate how the Auger spectrum has been manipulated. Simple application of iFFT results in a highly noisy spectrum, which can be partially alleviated by applying a noise removal filter. Yet, this operation is risky, as excessive noise rejection may lead to the suppression of crucial spectral features. An alternative to FFT and iFFT transforms is offered by the van-Cittert algorithm.

From the practical point of view, deconvolution algorithms find applications across various domains in experimental science [1,2,3,21,22,23]. As observed, they play a crucial role in enhancing resolution in imaging and spectroscopy, removing noise from signals, improving strain maps, and analyzing multilayer structures.

When measuring a physical observable, the response *h*(*t*) of an instrument to an input signal *f*(*t*) is determined by convolution Equation (2). Here, *g*(*x*) represents a signal component convolved with a second signal, which can be the impulse response of the instrument, spectral noise, or a spectral feature like the loss features mentioned earlier. The van Cittert algorithm for deconvolution is comprehensively detailed in [21,24,25]. For brevity, we will provide only a concise overview. The discrete form of Equation (2) is expressed as follows:(7)$$h(t)=\sum_{k=0}^{N-1}g\left(t-k\right)*f(t)$$
where t = 0, 1, … M + N − 1 being M and N, respectively, the number of data points of the vectors *f* and *g*. In our specific case, as noted above, *h*(*t*) represents the convolution of the Auger and loss feature spectra measured by the analyzer, *f*(*t*) the unaffected Auger spectrum, while *g*(*t*) describes the C 1s including the loss features. Equation (7) can be written in matrix form
h0h1h2........hM+N−1=g0000...0g1g000...0g2g1g0g0...0............gN−1gN−2gN−3gN−30gN−1gN−2gN−200gN−1gN−1.......gM+N−2.......gM+N−3.......gM+N−1M elementsf0f1f2...fM−1
or simply as
**h = G f**(8)

(bold characters represent the matrix form). If we multiply both sides by ***G^T^***, the result is the following:***G ^T^ h* = *G ^T^G f* or *f* = (*G ^T^ G*)^−1^*G^T^h***(9)

Here, ***G ^T^G*** is a Toeplitz matrix. We observe that the ***G ^T^G*** matrix is an M × M square matrix, where M is the dimension of the vector *f*. Also, ***G^T^h*** is a vector composed of M elements. The reconstruction of *f* poses an ill-conditioned problem, meaning that, irrespective of the solution method, a small relative error in the data ***h*** can lead to a substantial relative error in the computed solution. This is due to the matrix ***G ^T^G*** being nearly singular, and the direct inversion of ***G ^T^G*** for computing ***f*** does not yield a stable solution.

In such situations, regularization methods are typically employed, where the original functions are replaced by approximations which result in a solution less sensitive to errors in the data *h*(*t*). Various regularization algorithms have been proposed to solve Equation (8). Among them, the most popular ones include the Tikhonov–Miller regularization algorithm [26,27], the Riley algorithm [28], the Richardson–Lucy algorithm [29,30], and the van Cittert method [31,32]. The van Cittert procedure is grounded in an iterative process where [31,33]
*f ^n^*^+1^(*t*) = *f ^n^*(*t*) + [*h*(*t*) − *g*(*t*) ⨂ *f ^n^*(*t*)] (10)
where *n* represents the number of iterations. Considering Equation (8), Equation (10) can be rewritten as [21,24,25]
***f*** *^n^*^+1^ **=** ***f*** *^n^* **+** [***h*** **−** ***G f*** *^n^*] (11)
In the following we refer to the works of Xu et al. [24] and Morháč et al. [25].

From Equation (8), it follows that
***G^T^ G G^T^ h* = *G^T^ G G^T^ G f***(12)

Then, Equation (11) may then be rewritten as
***f*** 
*^n^*
^+1^ 
***= f*** 
*^n^ 
**+**  
*
**[*G^T^ G G^T^ h* − *G^T^ G G^T^ G f*** 
*^n^*
**]**
(13)

or more simply
***f*** *^n^*^+1^ **= *f***  *^n^* **+ [*h*′ − *G*′ *f***
*^n^***]**(14)
with ***h*′ *= G^T^ G G^T^ h, G*′ *= G^T^ G G^T^ G*.** To improve the speed of convergence was introduced a relaxation factor *μ* [34]:***f*** *^n^*^+1^  ***= f***
*^n^*  **+** *μ* **[*h*′ − *G*′ *f*** *^n^***]**
(15)

Convergence of the iteration is ensured if the diagonal elements of ***G*′** satisfy the following condition:(16)$${G\prime }_{ii}>\sum_{i=0,\;j\neq i}^{N-1}{G\prime }_{ji}\;\;\;i=0,1,\ldots ,N-1$$

The eigenvalues of the matrix ***G*′** are real. It is possible to demonstrate [25] that *μ* satisfy the conditions
0 < *μ* < 2/λ_max_
(17)
where λ_max_ represents the greatest eigenvalues of ***G*′**.

## 3. The *RxpsG* Software

*RxpsG* is a free software developed using the R libraries to analyze XPS spectra. R is an environment suitable for developing programs for statistical computing and graphics. It is a fully integrated and coherent system where all objects (functions and data) can be implemented. R includes a vast collection of libraries that enable the following:^◆^ an effective data handling and storage facility;^◆^ a comprehensive list of operators for the manipulation of arrays and matrices;^◆^ a large collection of libraries for mathematical and statistical analysis;^◆^ complete packages for plotting the analyzed data and saving graphical outputs in different file formats;^◆^ a well-consolidated programming language rich in instructions, enabling the construction of custom software.


*RxpsG* can be viewed as a collection of Graphical User Interfaces (GUIs) designed for performing spectral analyses on XPS data. It comprises a comprehensive set of tools for tasks such as reading spectra, performing background subtraction, and fitting peaks using appropriate functions. Additionally, it facilitates chemical speciation and elemental quantification. Special functions, including noise removal, differentiation, valence band analysis, and spectral convolution/deconvolution, are also implemented. *RxpsG* offers a range of options that make it easy to produce graphical outputs. Lastly, *RxpsG* provides a facility to summarize peak fitting results along with elemental quantification.

## 4. The Iterative Procedure in *RxpsG*

The deconvolution procedure outlined above was implemented by the authors of [19,20] to separate the loss features from the Auger spectrum of HOPG. Achieving accurate results necessitates the proper alignment of the C 1s and Auger spectra on the energy scale [20]. The above-mentioned study also highlights the importance of appropriate background subtraction. Nonetheless, it is noted that this step retains a degree of arbitrariness, as specific rules are not explicitly described. In the *RxpsG*, a free software developed in R^®^ [35], background subtraction can be achieved using different functions. For the extended C 1s core line which includes the complete loss feature region, a Tougaard background subtraction is suggested because it correctly describes the propagation of photoelectrons in the material starting from the Drude equation [36]. For the Auger spectrum, a Shirley background line shape was applied because it matches the spectrum edges as required by the authors of [20].

Despite these efforts, a common and significant challenge inherent in all these attempts stems from the convolution process. A first clue regards spectral alignment. In this work, we will use the loss feature excited by C 1s photoelectrons and the Auger spectrum obtained using the same spectrometer. Spectra are acquired on a highly oriented pyrolytic graphite exfoliated under vacuum to ensure a perfectly clean surface. C 1s and Auger spectra fall at rather different kinetic energies: the former is peaked at ~1202.2 eV, while the second falls in the binding energy range 280–240 eV. It is possible to align them on a common energy scale considering that the maximum of the possible kinetic energies of the Auger photoelectrons is obtained when the whole energy released by the relaxation of a valence electron into the C 1s hole (284.4 eV with our spectrometer) is absorbed by an electron at the Fermi level. This level in graphite corresponds to the top of the π band that falls at a null binding energy. As a consequence, the C 1s peak has to be placed at 284.4 eV on the kinetic energy scale of the Auger spectrum, as shown in Figure 1.

Another consideration regards the convolution process. If *m* is the number of elements of *f* and *n* is the number of elements of *g*, by definition, the convolution *h* is composed of *m + n* elements. However, by considering Equations (11) or (12), the value of *f ^n+^*^1^ (i.e., a function composed by *n* elements) is obtained through a calculation that involves a convolution (*n + m* elements). However, when we examine Equations (11) or (12), we find that the calculation for obtaining the value of *f ^n+^*^1^ (a function composed of *n* elements) involves a convolution with *n + m* elements. As previously mentioned, this issue can be addressed by multiplying the vector ***h*** and the matrix ***G*** with its transpose ***G^T^*** and applying Equations (11) or (12) when the relaxation factor *μ* is utilized. It is worth noting that attempts to use ***G^T^h*** as a reference to model the Auger spectrum in the van Cittert iterations have proven unsuccessful. The modeled function *f ^n^* has failed to accurately describe the original Auger spectrum. This suggests that ***G^T^h*** may not be a suitable reference for reproducing the Auger spectrum through van Cittert iterations. In *RxpsG*, the deconvolution option utilizes either the FFT or the van Cittert algorithm. Attempting to deconvolve the loss feature associated with the high-energy tail of the core lines from the corresponding Auger spectrum using FFT results in outcomes marked by unacceptable levels of noise, as anticipated. Conversely, the alternative van Cittert algorithm necessitates a procedure to reduce the sum of *m + n* points to *m*, as outlined in Equation (10). The flowchart of the deconvolution routine is shown in Figure 2.

In addressing this challenge, *RxpsG* retains *m* data corresponding to the convolution’s portion with a higher spectral power, while the remaining *n* − 1 points describing the featureless tail at a low kinetic energy are disregarded. This approach leverages a specific property of the convolution function. Given a generic function *f* with *m* elements and the Dirac function *δ* with *n* elements, it follows that
*f* ⨂ *g* = *f_c_*
(18)

Here, *f_c_* is equal to *f* plus *n* additional zeros introduced by the convolution with the delta function. In our case, the C 1s core line exhibits sharpness resembling a *δ* function, with the peak situated near the high-kinetic-energy edge. Consequently, the convolution of the Auger and the C 1s yields a spectrum consisting of a portion closely resembling the original Auger spectrum in the highly kinetic region and a featureless tail in the remaining low-energy region. This can be easily verified by checking the FWHM of C KVV ⨂ C 1s, which appears to be almost the same as that of the original C KVV spectrum. Then, C KVV ⨂ C 1s accumulates the information in a region with an extension substantially equal to that of the original Auger spectrum. This enables the selection of a spectral portion comprising *m* data points by aligning them with the original spectrum to minimize errors. The additional *n* points forming the tail can be neglected, as illustrated in Figure 3.

In Figure 3, truncating the featureless tail of the convolved spectrum introduces a discontinuity point because, at low kinetic energies, the edge is non-zero. The FFT transform of a transition to a non-null value in zero-time, such as in the case of a *δ* function or a square wave, contains all frequencies. For similar reasons, when computing the convolution using the van Cittert algorithm, it is evident that the discontinuity at the edge leads to spectral distortions.

In *RxpsG*, this issue is addressed by applying a damping factor to align the edges of the original and convolved spectra, as illustrated in Figure 4a,b. This damping factor serves to mitigate the spectral distortions arising from the discontinuity at the edge, ensuring a more accurate representation of the convolved spectrum without introducing artifacts.

Now, the iterative van Cittert algorithm can be applied. The signal-to-noise ratio (SNR) of the original spectra (Auger and C 1s in this example) may vary under different acquisition conditions. So, simple minimization of the standard deviation is insufficient to determine when to stop the iteration. Due to the low number of cycles required for convergence, the *RxpsG* software prompts users at each step to decide whether another iteration is needed to achieve a better result. Figure 5 illustrates the situation at the beginning and after the six iteration cycles sufficient to reproduce the original CKVV spectrum (black) using the convolution of the modeled CKVV with the C 1s (red spectrum). The green trace representing the spectral difference between the modeled and original spectra is approximately zero in the whole energy range.

Upon observing the deconvolved spectrum (depicted in blue), it becomes evident that the iterative process contributes to a rise in spectral noise. To address this issue, the software provides a *Denoise* option, aiming to control the level of noise in the deconvolved spectrum. Opting for this feature involves applying a slight shift either forward and backward of *μ*
**[*h*′ − *G*′ *f*** *^n^***]** with respect to ***f***  *^n^* (refer to Equation (13)) during each iteration. This procedure leads to noise reduction, resulting in an improved and acceptable spectrum.

Figure 5b,c display the deconvolution outcomes obtained without and with the application of the *Denoise* procedure, respectively. As is discernible, there is a noticeable reduction in the superimposed noise on the deconvolved spectrum (depicted in blue) in the second case, occurring just after six iterations. This option proves to be particularly effective, especially in scenarios where a higher number of cycles is necessary for convergence, leading to a substantial decrease in the noise level.

## 5. Algorithm Testing

The results of the proposed method were tested to assess the van Cittert algorithm’s ability to accurately deconvolve loss features from the Auger CKVV spectrum. The objective was to evaluate the outcome when applying the inverse operation to reproduce the original spectrum. To achieve this, we started with Equation (18), providing the deconvolved spectrum ***f*** *^n+^*^1^. In our example, ***f*** *^n+^*^1^ represents the deconvolved Auger spectrum at iteration *n* + 1. If ***f***  *^n+^*^1^ correctly describes the deconvolved CKVV spectrum, then ***f ^n+^*^1^** ⨂ ***g***, where *g* is the C 1s core line, must replicate the original Auger data. We compared ***f ^n+^*^1^** ⨂ ***g*** with the CKVV spectrum for different iteration numbers. The results are illustrated in Figure 6.

As evident from Figure 6a, increasing the number of iterations enhances the fidelity of reproducing the original spectrum. This trend is further illustrated in Figure 6b, where the root mean square deviation (RMSD) between the original Auger spectrum and the result obtained from the ***f ^n+^*^1^** ⨂ ***g*** convolution is plotted against the number of iterations. The RMSD decreases with each iteration, although the improvement in reproduction fidelity tends to plateau after the sixth iteration.

## 6. Conclusions

In conclusion, this discussion has focused on the deconvolution of loss features from the Auger spectra of carbon. Drawing from the past and present literature, guidance is provided for the correct pre-processing of the original data, specifically emphasizing accurate background subtraction and energy alignment of the spectra. The deconvolved spectrum is obtained by applying the van Cittert algorithm. However, the direct application of the iterative procedure requires caution due to the mismatch of the number of elements between convolution and the original spectrum in the van Cittert algorithm. Reproduction tests have confirmed that *RxpsG* provides a reliable method for accurately calculating the deconvolved spectrum. This involves selecting the appropriate data segment and utilizing a denoising option to effectively minimize noise in the resultant spectrum.

## Figures and Tables

**Figure 1 materials-17-00763-f001:**
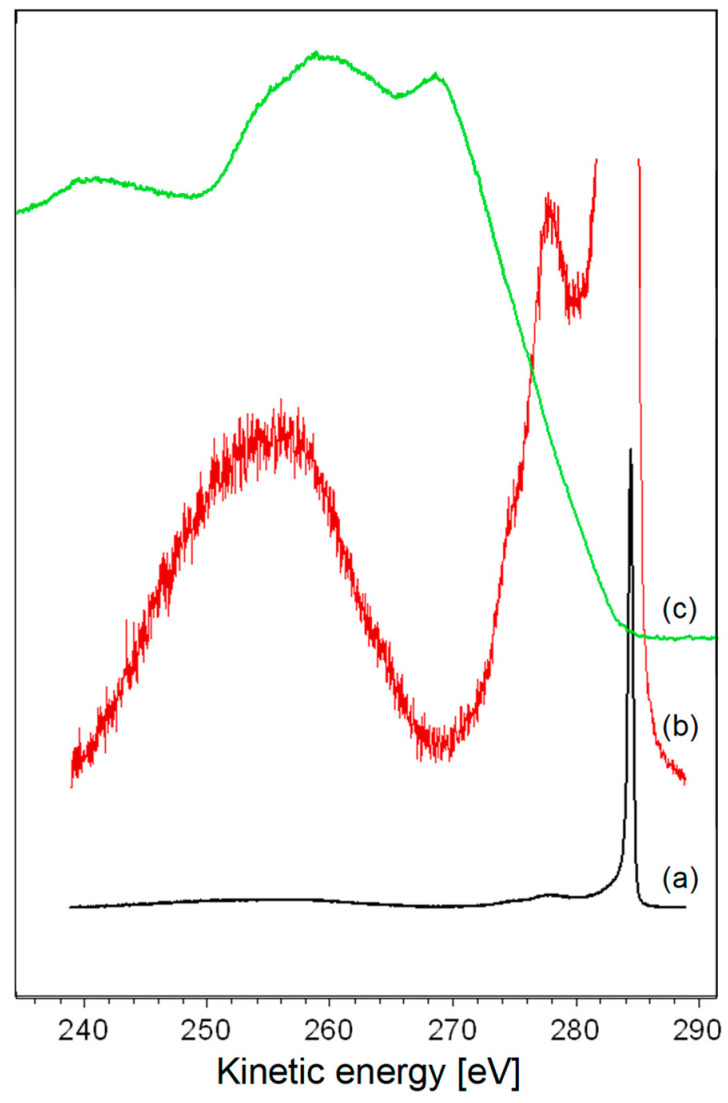
(**a**) Position of the core and loss features aligned on the kinetic energy scale of the Auger spectrum; (**b**) amplified view of the C 1s loss feature; and (**c**) Auger spectrum.

**Figure 2 materials-17-00763-f002:**
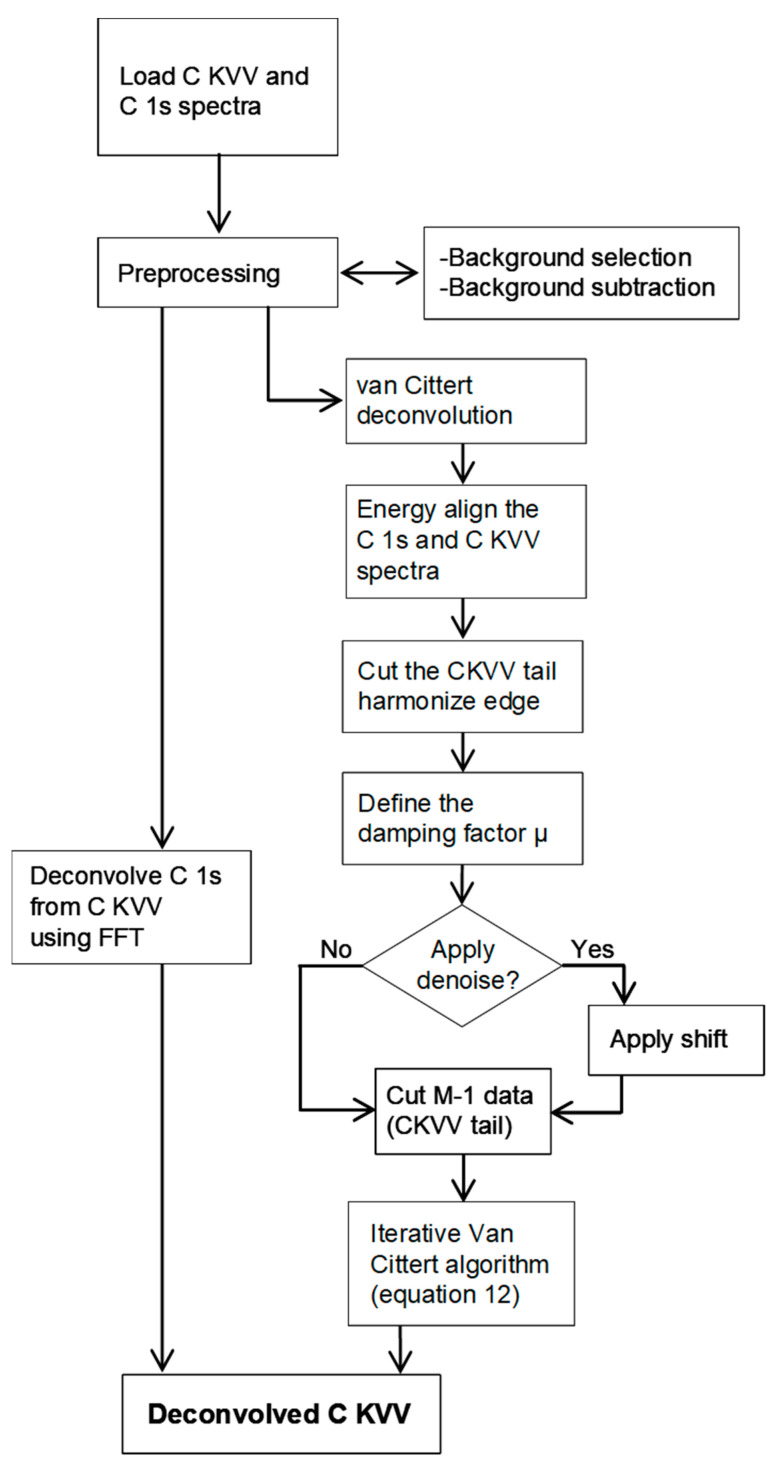
Flowchart of the deconvolution routine. For clarity, the scheme refers to CKVV and C 1s spectra.

**Figure 3 materials-17-00763-f003:**
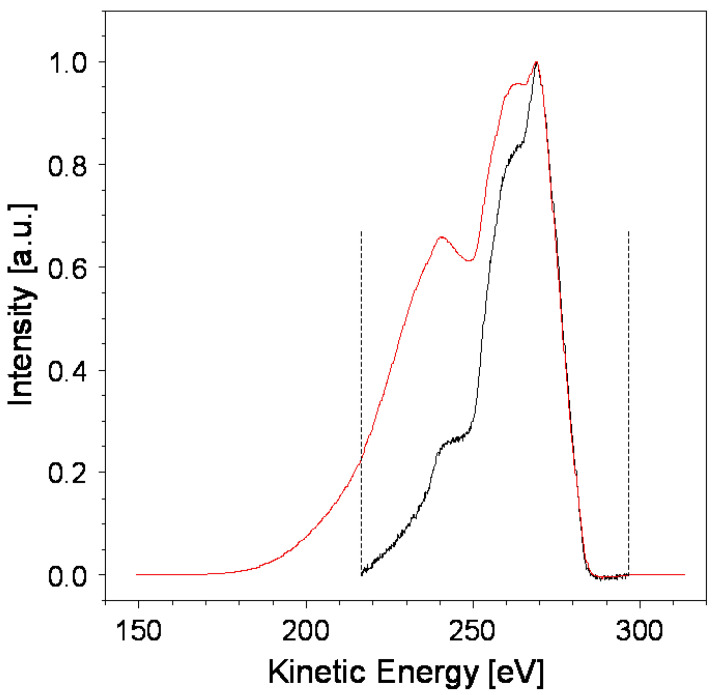
Original background-subtracted Auger spectrum (black) and the C KVV ⨂ C 1s convolution (red). The dashed line indicates the point where the convolution will be truncated. The dashed lines delimit the portion of the spectrum used for the deconvolution. In particular, at a low kinetic energy the featureless tail is disregarded during the van Cittert iteration process.

**Figure 4 materials-17-00763-f004:**
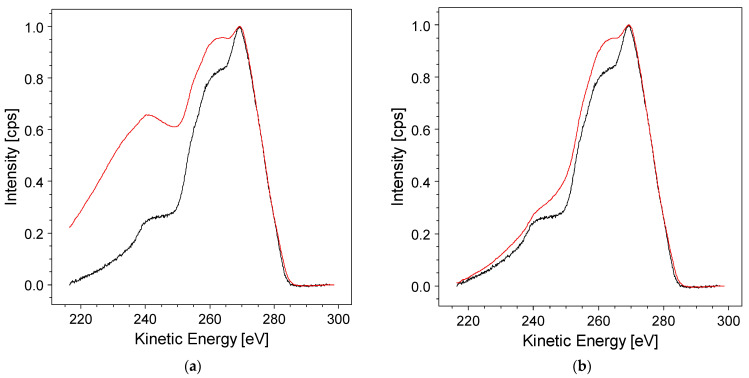
Applying a damping factor to harmonize the low-kinetic-energy (KE) tail of the convolved spectrum from the van Cittert algorithm (in red) with the original background-subtracted Auger spectrum (in black). Panels show results for different damping factors: (**a**) damping factor = 0.1; and (**b**) damping factor = 0.8.

**Figure 5 materials-17-00763-f005:**
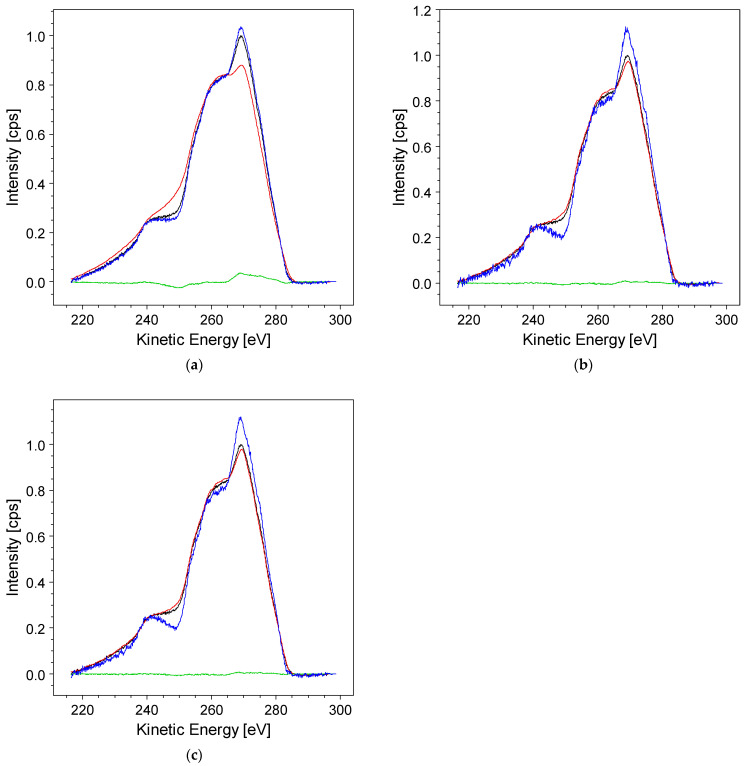
The result of the *Van Cittert* iteration at the beginning, with the original data (black), ***G*′ *f*** *^n^* convolution (red) iteration = 1 in (**a**), iteration = 6 in (**b**,**c**), the deconvolved spectrum (blue), and the difference ***h*′ − *G*′ *f*** *^n^* (green); (**b**) after six iterations, the convolution overlaps the original data, and the difference is almost null in the whole energy range; (**c**) when the *Denoise* option is selected, a less noisy deconvolved spectrum (blue) is obtained.

**Figure 6 materials-17-00763-f006:**
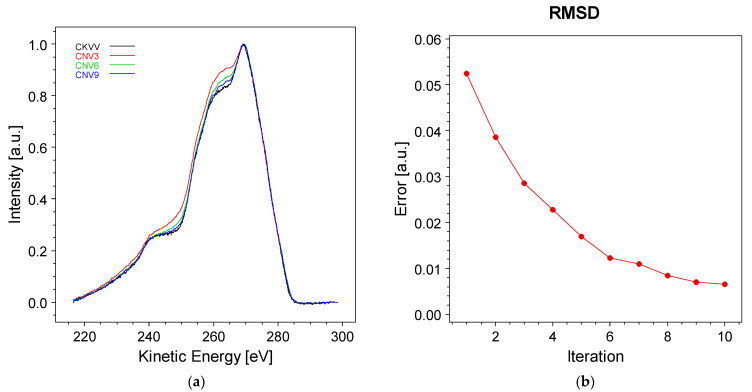
(**a**) Comparison of the original Auger CKVV spectrum with the convolution result ***f ^n+^*^1^** ⨂ ***g***, ***f ^n^*****^+1^** representing the output of the van Cittert method and ***g*** the C 1s spectrum, at iterations three, six, and nine; (**b**) root mean square difference (RMSD) between the original Auger spectrum and the convolution result ***f ^n+^*^1^** ⨂ ***g*** as a function of the iteration number.

## Data Availability

Data are available on request using the corresponding author e-mail address.

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
