# Peer review of "Application of the Van Cittert Algorithm for Deconvolving Loss Features in X-ray Photoelectron Spectroscopy Spectra"

_materials, 2024, doi:10.3390/ma17030763_

Round 1

Reviewer 1 Report

Comments and Suggestions for Authors

The manuscript is a short overview of the current state of art, focusing on the analysis of Auger spectra which is obtained through XPS.  

Reviewer comments

 Auger spectra obtained via XPS is, usually, difficult to analyse although their presence can be important as complement to XPS results, giving/add more, and maybe clear, information about the system under analysis.

In that sense, this manuscript gives a good example of the importance of deconvolution and present the advantages applying the van Cittert algorithm.

As for my knowledge, this software and algorithm is not well known. Therefore, due to its importance, it is suggested to accept the manuscript for publication.

Author Response

//

Reviewer 2 Report

Comments and Suggestions for Authors

The author reports the application of van Cittert algorithm in his authored open source software RxpsG, taking deconvolution of loss features from carbon's Auger spectra as an example. It should be helpful in the relevant research area, and thus is worth known by the community.

Misspelling and some minor grammatic error have been found. Correct ones should be: line 112, “… this approach…”. Line 185, “This level in graphite corresponds to …”.

Author Response

//

Reviewer 3 Report

Comments and Suggestions for Authors

Author Response

//

Reviewer 4 Report

Comments and Suggestions for Authors

1. The theoretical framework of inverse problem for solving Equation 7 is well-established. Why not directly incorporate the regularization operator into Equation 8?

2. According to the conventions of mathematical inverse problem theory, it is unnecessary to define the Toeplitz matrix for the convenience of readers.

3. What is the mathematical basis for the derivation from Equation 7 to Equation 9 or 10? I am confused as Equation 7 does not seem to lead to Equation 9 or 10, considering both the Newton gradient and Taylor series perspectives. The mathematical relationship between Equation 7 and either Equation 9 or 10 is not apparent; In other words, the algorithm lacks a mathematical foundation.

4. How is the relaxation factor μ determined? Will μ function as a software's slider control, impacting the accuracy of the images (including shape, peak size, smoothness, etc.)? The determination of the optimal relaxation factor μ has well-established theoretical references in mathematics of inverse problem fields.

5. There are numerous lines in Figures 2-4, and their specific meanings are unclear. Annotations are required for clarity.

6. How is the accuracy of the algorithm verified? A numeric test is essential, adhering strictly to the design of the inverse problem algorithm. Initially, a test problem (with specified real results) must be formulated, and then, utilizing the information from the test problem, algorithm recovered results are reconstructed and compared with the real results.

Comments on the Quality of English Language

Enhance the readability of sentences when describing the images.

Author Response

//

Round 2

Reviewer 3 Report

Comments and Suggestions for Authors

Quality of the updated manuscript is significantly improved and meet the standard of the journal publication. 

Reviewer 4 Report

Comments and Suggestions for Authors

it can be accepted